# Combining country indicators and individual variables to predict soil-transmitted helminth infections among migrant populations: A case study from southern Italy

Jana Purkiss[1]*, Paola Pepe[2], Naím Alex Karol Poplawski[3], Maria Paola Maurelli[2], Luciano Gualdieri[2], Laura Rinaldi[2], Emanuele Giorgi[1]

**1** Centre for Health Informatics, Computation and Statistics (CHICAS), WHO collaborating centre, Lancaster University Medical School, Lancaster University, Lancaster, United Kingdom, **2** Department of Veterinary Medicine and Animal Production, University of Naples Federico II, CREMOPAR, WHO Collaborating Centre ITA-116, Naples, Italy, **3** Institut für Tropenmedizin, Reisemedizin und Human parasitologie, Eberhard Karls Universität Tübingen, Tübingen, Germany

* j.purkiss@lancaster.ac.uk

## Abstract

An increase in global migration towards developed countries along with climate change has led to the occurrence of Neglected Tropical Diseases (NTDs) in otherwise non-endemic countries. In this paper we focus on Soil Transmitted Helminth (STH) infections which disproportionately affect people living in poverty in tropical regions. To reduce the threat of STHs in migrant populations living in non-endemic countries, diagnosis and treatment are paramount but also present logistical challenges. This study investigates how statistical modelling can be used to assist the identification of individuals infected with STHs. Specifically, we show how to combine individual variables (e.g., age, sex and time in Italy) with publicly available country indicators (Human Development Index, Multidimensional Poverty Index and Inequality-adjusted Human Development Index) which describe development in the migrant's country of origin. We combine these indices and their factors in binomial mixed-effects models which can be used to predict the status of STH infections in migrant populations. By presenting a case study on migrants in southern Italy, we assess the relative importance of the individual-level variables and country-level indicators in enhancing the predictive power of the models. The results show that the country-level indices play a more important role but also highlight that individual data can help improve the model performance when combined with the former. To the best of our knowledge this is the first study investigating using country-level indicators to predict parasite infection status of migrants. Our study indicates that statistical models can play an important role in reducing the resources required to identify migrants requiring anthelmintic treatment against STHs and help to make statistically informed decisions.

**Data availability statement:** The data used for the analyses in this paper are owned by WHO Collaborating Centre for Diagnosis of Intestinal Helminths and Protozoa (WHO CC ITA-116) in which the original surveys were conducted. Permission to use the data for our analyses was obtained via formal agreements between WHO CC ITA-116 and the involved universities and organisations; all data were de-identified before use. Researchers are welcome to access this data via Zenodo (https://doi.org/10.5281/zenodo.15210604)

**Funding:** This work was supported by the National Institute of Health Research (NIHR302758 to JP). The funder had no role in study design, data collection and analysis, decision to publish, or preparation of the manuscript.

**Competing interests:** The authors have declared that no competing interests exist

## Author summary

Neglected tropical diseases disproportionately affect impoverished populations, primarily in tropical regions. With increasing global migration, individuals infected with neglected tropical diseases may move to countries where these diseases are not typically found, making early diagnosis and treatment essential. We demonstrate how statistical models can be used to aid the identification of people who may be infected with these parasitic diseases. Our focus is on showing how publicly available information on the country of origin of migrants, can be combined with individual-level information collected from screening centres, to improve the predictive performance in the identification of infected cases. We look at a specific neglected tropical disease known as soil-transmitted helminth infections and use a case-study of migrants in the Campania region, southern Italy to conduct our investigation.

## Introduction

Soil Transmitted Helminth (STH) infections are one of twenty-one Neglected Tropical Diseases (NTDs) identified by the World Health Organisation (WHO) as disproportionately affecting people living in poverty [1]. These infections are caused by a group of intestinal parasites (*Ascaris lumbricoides*, *Trichuris trichiura*, the hookworms *Ancylostoma duodenale* and *Necator americanus,* and *Strongyloides stercoralis*) spread via their eggs or infective larvae in human stool which can contaminate soil in areas of low sanitation [2]. Over 1 billion people are estimated to be infected with STHs, with most of those living in tropical regions in low- and middle-income countries [3] and the disability-adjusted life years (DALYs) due to STHs is estimated at 1.9 million [2].

As a result of the current international migration patterns, along with climate change and international travel, these conditions have become more frequent in recent decades in non- endemic areas [4,5].

Consequently, NTDs, particularly STHs, could be a pressing concern for public health officials in developed countries [6–11] because these diseases may pose unfamiliar scenarios for healthcare services in developed nations, with infections from NTDs often being underdiagnosed, diagnosed belatedly, or inadequately managed [12]. Some STHs, e.g., *T. trichiura* and hookworms have a long lifespan in the human body and can be responsible for major disabilities when not diagnosed [13,14]. Several screening protocols have been suggested to control NTDs among migrant populations [6,15,16] however, most European countries have no systematic mandatory regulation regarding reporting and surveillance of parasitic NTDs [12]. Screening can be carried out in two different ways: panel screening in which diagnosis is universal regardless of country of origin, and targeted screening in which diagnosis is done based upon a risk-assessment [15]. For targeted screening to be effective, robust statistical methods that make best use of available data can help best

identify individuals to screen for diagnosis. In this paper we propose a model-based approach to identify which migrants may be most likely to carry STH infections using both individual variables and information about the country of origin. A model-based approach, such as the one outlined in this paper, might provide an effective data-driven approach to inform targeted screening which can help to reduce the burden placed on specialist parasitology laboratories [15].

Predictive models can be developed using both individual-level variables collected directly from the migrants and country-level indicators which can be obtained from publicly available sources. Since migrants from the same country might experience comparable levels of exposure, in lack of more detailed spatially referenced information, country-level indicators may help to account for this and improve the accuracy of a predictive model [17]. Nevertheless, relying solely on country-level indicators may prove insufficient in capturing all the variables that contribute to an individual's susceptibility to STHs in their country of origin. Mixed-effects models can help to alleviate this issue using so-called random effects. The use of both individual-level variables and country-level indicators helps to account for sources of heterogeneity in the data such as: biased age and sex distributions, under-represented countries, and different prevalence rates among the STH species [18].

One example of a widely used country-level index to quantify the overall development of a country is the Human Development Index (HDI). Proposed by the United Nations Development Programme (UNDP) in 1990, HDI is used as a composite measure encompassing health, living standards, and education (see S1 Fig) [19]. HDI has been linked with a wide range of diseases: cancer [20], tuberculosis [21], hypertension [22], Covid-19 [22], asthma and pneumoconiosis [23], and coinfection between intestinal parasites and *Helicobacter pylori* [24]. Previous studies have examined the link between HDI and infection with parasitic infections [25], although the results obtained from the first study, conducted in Peru between 2012 and 2016, concluded that there was no association between the prevalence of intestinal parasites and HDI at a sub-national level [26]. In response to this result, the individual components of the HDI and additional country characteristics, such as a 'Worm Index' [27], could be used in predicting STH infection status. Such country-level indices are publicly available and can be easily obtained from online sources. The Worm Index, proposed by Hotez and Herricks [27], quantifies the burden of STHs in a country by combining the number of school-aged children requiring treatment for STHs and schistosomiasis with the number of people requiring treatment for lymphatic filariasis and dividing by the total population. Hotez and Herricks [27] found a statistically significant inverse relationship between HDI and Worm Index, however, they also found that India, Indonesia and the Philippines have a relatively high Worm Index despite being countries categorized as high HDI nations by the UNDP [27]. This paradoxical relationship in some countries prompted us to investigate a Worm Index could be used as a predictor of STH infections among migrants. Whilst there has been a previous study which utilized information about migrants' countries of origin to predict tuberculosis [21], no studies have made use of this information to predict the likelihood of STH infections among migrants.

In this paper we illustrate the proposed framework using a case study of STH infections among a subgroup of the migrant population of southern Italy, the sixth most popular migrant destination country in Europe in 2020, with immigrants making up 10.1% of the total population [28]. The overall aim is to assess the advantages and limitations of country-level indicators and individual-level variables in developing predictive models for STH infections among such migrant populations. To achieve this, cross-validation techniques are used to compare a series of models which are informed by either country-level, individual-level, or by both sources of information. The generalizability of this approach to other geographical contexts is then discussed.

## Materials and methods

### Ethics statement

Ethics approval was obtained from the Faculty of Health and Medicine Research Ethics Committee, Lancaster University for the analysis of this data. FHM-2024–4877-DataOnly-1.

## Study area and data collection

Migrants who arrived in the Campania region (southern Italy) were randomly screened for intestinal parasites. The spokes for the stool collection were set in different migrant assistance centers in the region: (i) two Medical Centres for the Health Protection of migrants (ASL-NA1 and ASL-SA) between 2006 and 2020 and (ii) eight Centres for extraordinary reception (CAS) between 2022 and 2023.

At the time of stool collection, patients had not taken anthelmintic drugs for at least one year, as these could have interfered with the laboratory results. The samples were collected after the signature of an informed consent document by the participants indicating that they understood the purpose and the procedures required for the study and that they were willing to let their child participate in the study (in the case of parents of children). Furthermore, patients were asked to fill in a questionnaire containing their country of origin, age, sex and the length of time they had been in Italy. All data records were entered and analysed using Microsoft office excels worksheet. Patients' records with incomplete information were not included in the analysis.

## Parasitological examinations

For each patient, one stool sample was analysed. The stool samples were sent, within 24 hours, to the laboratories of the WHO Collaborating Centre for diagnosis of intestinal helminths and protozoa (WHO CC ITA-116, University of Naples Federico II, Italy). An aliquot of two grams of stool sample was analysed by the FLOTAC dual technique [29], using two flotation solutions (FS), i.e., FS2 (sodium chloride-based; specific gravity, s.g. = 1200) and FS7 (zinc sulphate-based; s.g. = 1350), according to the protocol described by Maurelli et al. [30].

## Data processing

Country-level indicators, Human Development Index (HDI), Multidimensional Poverty Index (MPI) and Inequality-adjusted Human Development Index (IHDI), matching the countries given by the migrants were obtained from publicly available sources (https://hdr.undp.org/). The components that constitute these indices are: life expectancy in years, mean of years of schooling for adults aged 25 years and more and expected years of schooling for children of school entering age, gross national income (GNI) in USD per Capita. Other factors utilized in our analysis are: population size in millions, population density in people per sq km of land (https://data.worldbank.org), percentage of population using basic sanitation services, by location, and percentage of population using basic drinking water services, by location (https://www.unwater.org/). Since some countries did not have reported indices for specific years, the indices used in the analysis were subset to include HDI, Life Expectancy, GNI per Capita, Population, and the percentages of the population using basic sanitation and drinking water services. No data was available for 2023, so the indices from 2022 were used in its place.

The individual variables were referred to as age, gender, and time in Italy (calculated as the difference between the last arrival in Italy and the date of parasitological analysis). Fig 1 summarizes the country-level variable selection process in which the indicators were grouped into four domains: Developmental, Financial, Demographic and Water, Sanitation and Hygiene (WASH). For each domain one variable was selected such that there was no correlation greater in magnitude than 0.7 with any variable from another category to reduce collinearity [31] yet that had the highest possible correlation with the variables in the same category so it best represented that domain. The selected variables were: Life Expectancy in years, GNI per Capita, Population, and the percentage of population using basic sanitation services. From here on, these variables will be referred to as Life Expectancy, GNI per Capita, Population, and Sanitation, respectively.

Rows of data which were missing data for individual-level variables or country-level indicators were removed from the data.

Kang et al. [32] calculated a Worm Index based on two different data sources; data from the WHO and data from the Institute for Health Metrics and Evaluation (IHME) for the Global Burden of Disease Study (GBD). From here on, these will be referred to as the WHO Worm Index and the GBD Worm Index, or collectively as the worm indices. For each data

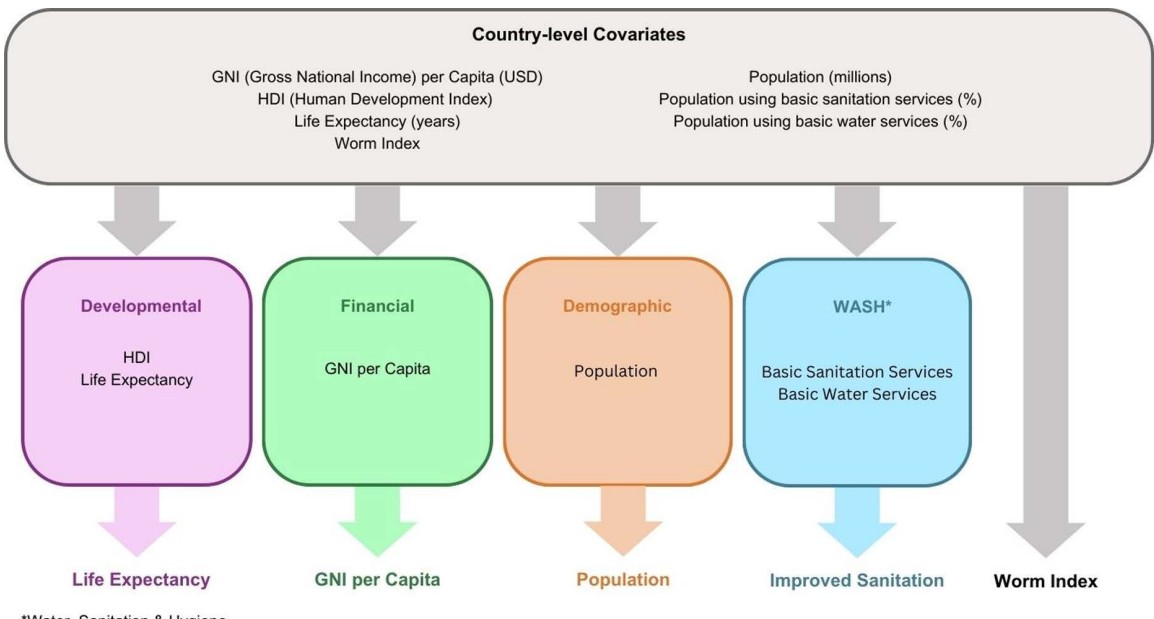

**Fig 1. The country-level variable selection process.** Country-level indicators are grouped into domains (reflecting the HDI domains where possible [19]): Developmental, Financial, Demographic, and Water Sanitation and Hygiene (WASH). One variable from each domain is selected such that it has the highest possible correlation with other covariates in that domain but has no correlation greater in size than 0.7 with any other domain. The Worm Indices [32] were also obtained if they were published for the country in question.

source, Kang et al. [28] published the top 100 countries' worm indices by rank. 46 (42) out of the 64 countries in our dataset were within the top 100 countries ordered by WHO (GBD) Worm Index. Given this lack of information and the potential impact on the results of this study, we run three analyses, firstly not considering the worm indices, then considering if each country is in the top 100 countries ordered by the worm indices, and finally on a subset of the data for which the worm indices are available by considering the rank of the worm index. This is done based on the WHO Worm Index and the GBD Worm Index separately. The efficacy of these models in prediction are only compared for modelling overall STH infection rather than for each species individually and only the most informative model found in the first analysis is extended.

All analysis and data visualisation were carried out in R [33], using packages boot [34], countrycode [35], forestplot [36], dplyr [37], ggplot2 [38], gridExtra [39], lme4 [40], matrixStats [41], patchwork [42], pROC [43], raster [44], rnaturale-arth [45],table.glue [46], tidyr [47], and viridis [48].

## Model formulation and parameter estimation

We compare three different Binomial mixed models, making use of either individual-level variables, country-level indicators or both sources of information. Below, we provide an overview of these models using a symbolic representation. For more technical details and formal explanations, we refer the reader to S1 File.

1. The first model (M1) accounts for the individual-level variables and an additional term accounting for the variance between individuals which could not be accounted for by the covariates and will be referred to as the individual random effect. The symbolic representation of this model is:

$$\text{Log-odds of individual's STH likelihood} = I + \alpha_1 + \alpha_2 + \alpha_3 + \text{Individual random effects} \qquad (1)$$

Where all the covariates are individual variables provided by the migrants. Specifically, $\alpha_1$ is their age, $\alpha_2$ is their sex, and $\alpha_3$ is their time spent in Italy. The individual random effects are assumed to be independent, identically distributed zero-mean Gaussian variables.

2. The second model (M2) accounts for the country-level indicators and an additional term accounting for the variance between countries which could not be accounted for by the covariates which will be referred to as the country random effects. The symbolic representation of this model is:

$$\text{Log odds of individual's STH likelihood} = I + \beta_1 + \beta_2 + \beta_3 + \beta_4 + \text{Country random effects} \tag{2}$$

Where all the covariates are indicators associated with the individual's country of origin. Specifically, $\beta_1$ is the life expectancy, $\beta_2$ is the GNI per Capita, $\beta_3$ is the population, and $\beta_4$ is the sanitation. The country random effects are assumed to be independent, identically distributed zero-mean Gaussian variables.

3. The third model (M3) accounts for both individual-level variables and country-level indicators, along with an individual random effects term and a country random effects term. The symbolic representation of the model is:

$$\text{Log odds of individual's STH likelihood} = I + \alpha_1 + \alpha_2 + \alpha_3 + \beta_1 + \beta_2 + \beta_3 + \beta_4$$
$$+ \text{Individual random effects} + \text{Country random effects} \tag{3}$$

where the covariates and random effects terms are as defined in (1) and (2).

In all models, $I$, the intercept term, and the covariates vary by species. All the continuous variables in the models are scaled by subtracting the mean and dividing by the standard deviation.

A two-step estimation procedure was used to fit each of the mixed effects models. This is demonstrated using M3 but the method was the same for each model with the relevant terms left out as indicated in (1) and (2). First, only the fixed effects were included as covariates in a standard generalized linear model using iterative least squares estimation. The symbolic representation of this is:

$$\text{Log odds of individual's STH likelihood} = I + \alpha_1 + \alpha_2 + \alpha_3 + \beta_1 + \beta_2 + \beta_3 + \beta_4 \tag{4}$$

The fitted values, on the log-odds scale, were then used as an offset in a mixed effects model including the random effects terms. The symbolic representation of this is:

$$\text{Log odds of individual's STH likelihood} = I + \hat{\alpha_1} + \hat{\alpha_2} + \hat{\alpha_3} + \hat{\beta_1} + \hat{\beta_2} + \hat{\beta_3} + \hat{\beta_4}$$
$$+ \text{Individual random effects} + \text{Country random effects} \tag{5}$$

Where, for example, $\hat{\alpha_1}$ represents the value of $\alpha_1$ obtained in (4). Fitting the model using an offset was done to simplify the estimation of the random effects model and alleviate the computational burden. To generate confidence intervals for the regression coefficients that acknowledge the overdispersion of the data, we developed the following bootstrap procedure:

1. For each individual observation in the original dataset, two gaussian distributions with mean 0 and variances set to the values estimated in (5) were sampled to obtain a country random effect and an individual random effect. The fitted value of the log-odds for each individual based only upon the fixed effects was also calculated. The simulated random effects and the fitted values were added for each individual and used as the parameter to sample from a binomial distribution to simulate a new piece of data.

2. Fit model (4) to the simulated data and record the estimated regression coefficients. If the model did not successfully converge, the coefficients were not recorded.

3. Repeat the above steps 1000 times dismissing any iterations in which the coefficients did not converge.

4. Compute the 95% confidence intervals for the regression coefficients, using the 2.5% and 97.5% quantiles from the 1000 values obtained from the previous step.

## Assessment of predictive performance

The predictive power of M1, M2, and M3 were assessed in two main scenarios: predicting for a new individual from a country in the existing dataset and predicting for a new individual from a new unsampled country. In each of the main scenarios, we assess the predictive power for predicting infection likelihood for any STH infection and for infections of each of the three species separately. The predictive power is assessed using a cross-validation technique outlined below.

1. Data from countries that had only one individual were excluded since it would not be possible to represent the country in both the training set and the testing set.

2. One individual from each country was selected at random to be included in the test set and the remaining data formed the training set. This was done so that the model could be tested for each country in the dataset, whilst keeping the training set as large as possible to minimize convergence issues.

3. The training data was used to fit the mixed-effects models using the two-step estimation procedure outlined above.

4. Predictions for the test set were made for each prediction scenario, where for the second scenario, the name of the country the individual was from was first removed. These predictions were recorded.

5. Steps 1–6 were repeated 1000 times.

6. Receiver Operator Curves (ROCs) [24] were plotted for each prediction scenario, and the area under the curves (AUCs) calculated.

7. The models were compared using the ROCs and AUCs for each prediction scenario, where the closer the AUC value is to 1, the better the predictive power of the model. The DeLong 95% CI were found for each AUC.

8. The values of the AUCs were compared for statistical difference using the DeLong method [25] where the null hypothesis was that two AUC values were equal.

The DeLong confidence intervals and test utilizes the realization of the Mann-Whitney statistic and can be used as an estimate of the AUC. This test is used to test the null hypothesis that two AUCs are the same, or in other words that the two models under consideration have the same predictive power based on the AUC. For more details, see S2 File.

## Extension of model using worm indices

Given its predictive performance compared to the other models, M3 was extended to include information about the worm indices.

The two new models $M4_{WHO}$ and $M4_{GBD}$ have the symbolic representation:

$$\text{Log odds of individual's STH likelihood} = I + \alpha_1 + \alpha_2 + \alpha_3 + \beta_1 + \beta_2 + \beta_3 + \beta_4 + W_{WHO}$$
$$+\text{Individual random effects + Country random effects} \tag{6}$$

and

$$\text{Log odds of individual's STH likelihood} = I + \alpha_1 + \alpha_2 + \alpha_3 + \beta_1 + \beta_2 + \beta_3 + \beta_4 + W_{GBD}$$
$$+ \text{Individual random effects + Country random effects} \tag{7}$$

where the covariate and random effect terms are defined as in (3) apart from $W_{WHO}$ and $W_{GBD}$ which take the value 1 when the individual's country of origin is included in the list of the top 100 countries ordered by WHO Worm Index and GBD Worm Index data, respectively.

Two further models were fit to subsets of the original data. Two subsets were created; one including only the countries for which the WHO Worm Index was available and another including only the countries for which the GBD Worm Index was available. Firstly, M3 was refitted to both these subsets, then the two additional models, $M5_{WHO}$ and $M5_{GBD}$ were fitted to their respective subset of data. The symbolic representation of the models are as in (6) and (7) with the exception that $W_{WHO}$ and $W_{GBD}$ now take the value of the Worm Indices.

The methodology for parameter estimation and the assessment of predictive performance for these new models are as above, but the predictive performance will only be assessed for overall STH infections rather than for each species individually.

All results are given to 3 significant Figs and significance level is taken to be 0.05.

## Results

### Data summary

The data consists of 3,830 individuals (migrants) from 64 countries. Of these, 87.4% were males. Fig 2 shows the number of individuals from each country. We observe that 12 countries were only represented by one migrant. Bangladesh, with

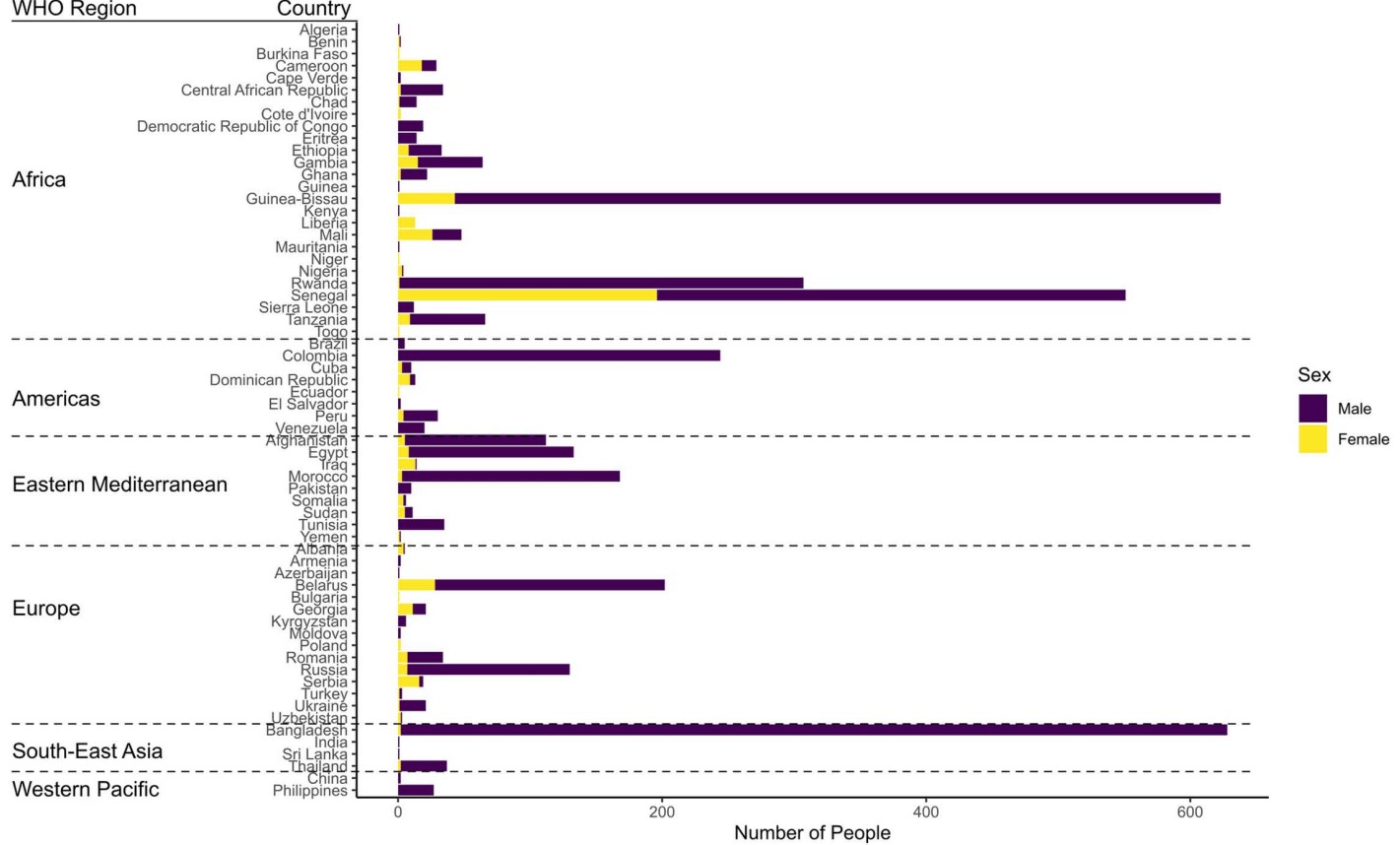

**Fig 2. A graph to show the number of migrants originating from each country in the dataset.** The proportion of males and females represented in each country of the dataset are shown in purple and yellow respectively. The countries are grouped into the WHO regions which is shown on the left-hand side of the chart.

628 migrants, was the most represented country in the dataset. The distribution of the countries according to WHO regions was as follows: 26 in Africa (AFRO), 8 in the Americas (PAHO), 9 in Eastern Mediterranean (EMRO), 15 in Europe (EURO), 4 in South-East Asia (SEARO), and 2 in Western Pacific (WPRO). Fig 2 also shows the gender distribution by country, from which we notice a high level of heterogeneity across countries, for instance, all the individuals from Mali were female, all of the individuals from Peru were male, and many countries were represented by both male and female migrants of differing proportions. Fig 3 shows the distribution of the continuous covariates used in the models. The age distribution is right skewed where 90% of the individuals were younger than 41, and the median age is 27. Fig 3 also shows that 67.1% of people have been in Italy for less than 2 years, and 60.6% of those individuals have been in Italy less than 6 months. The median values of the country-level covariates: life expectancy, GNI per Capita, population, and sanitation are 63.1, $3970, 26.8 million, and 44.1% respectively. Fig 4 shows the prevalence of STH infections in the dataset aggregated by country showing higher prevalence in Africa and South-East Asia than in other WHO regions. In Africa, the country of origin with the highest prevalence is with Guinea Bissau with 25% STH prevalence among migrants. In South-East Asia, the country of origin with the highest prevalence is Bangladesh with 18.6% STH prevalence among migrants. Additional plots, showing the prevalence of STH infections for each species separately, are shown in S2 Fig. The overall STH prevalence was 10.1%, specifically 0.4% for *A. lumbricoides*, 6.4% for hookworms, and 3.3% for *T. trichiura*. Three people in the study had a

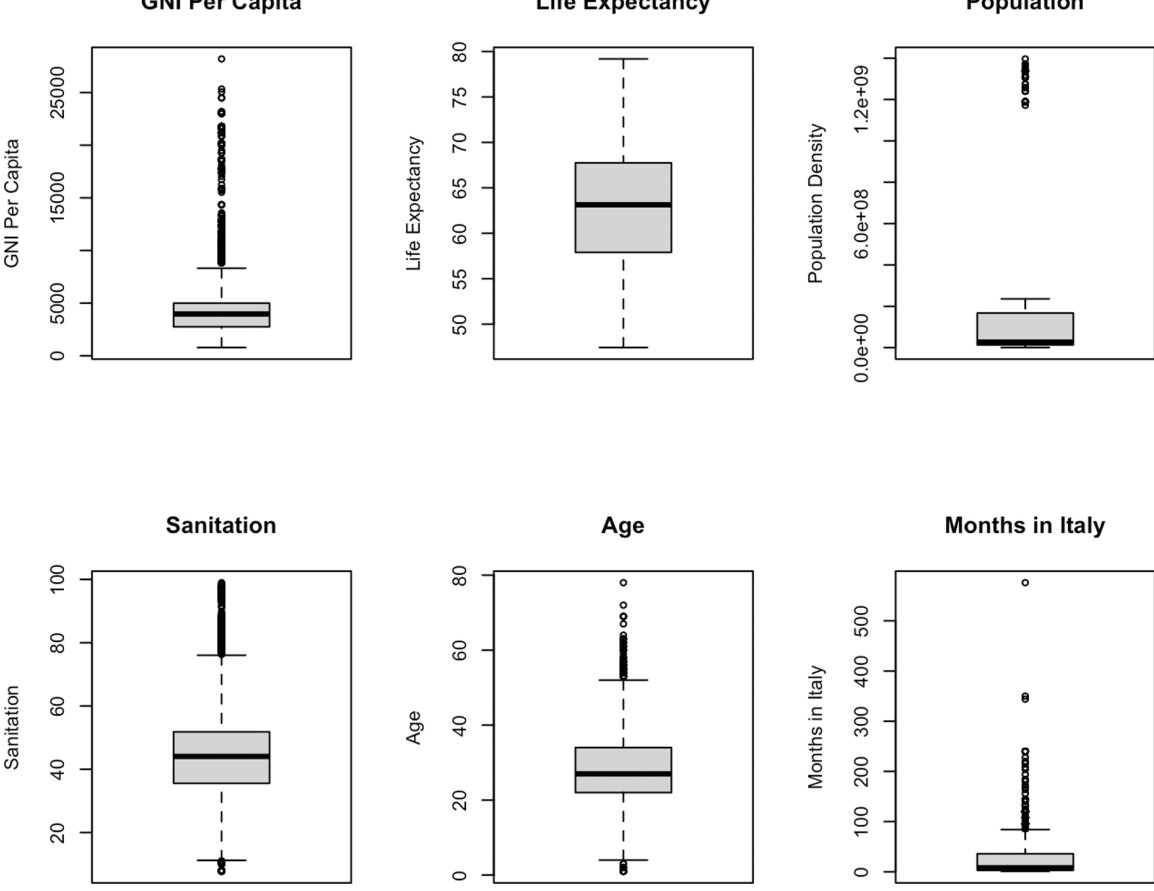

**Fig 3. Boxplots to show the distribution of the continuous covariates used within the models.**

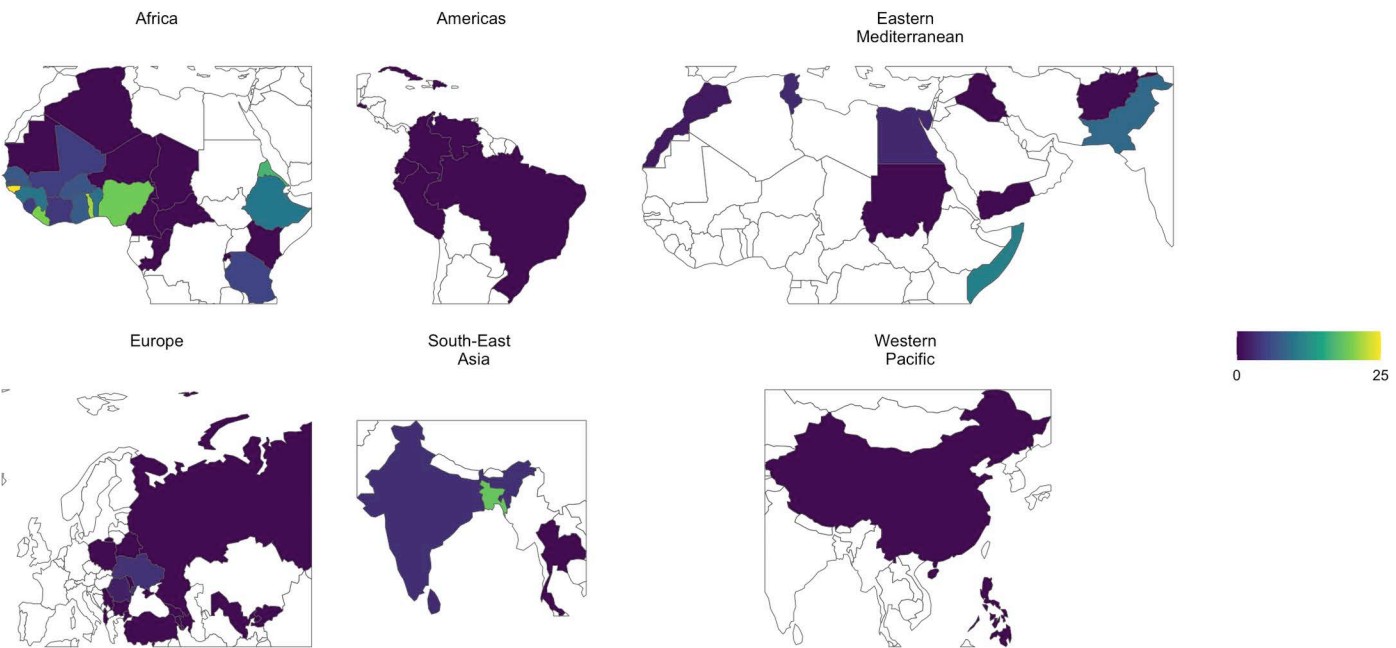

**Fig 4. A set of maps to show the prevalence (%) of STH infections of migrants by their country of origin split by WHO region.** Made with Natural Earth.

coinfection with *A. lumbricoides* and *T. trichiura*, 3 had a coinfection with *A. lumbricoides* and hookworms, 15 had a coinfection with *T. trichiura* and hookworms, nobody had a coinfection where all three STHs were present.

267 individuals were sampled at the centres for extraordinary reception, compared to 3,563 individuals who were sampled at the Medical Centres for the Health Protection of migrants. Pearson's Chi-squared tests with Yate's continuity correction [49] were performed to determine if the prevalence of all STH infections, hookworm infections, and *T. trichiura* infections were statistically different between these two groups of migrants. The p-values for all these tests were >0.05 (0.377, 0.241, 1.00) and so we assumed in this analysis that the risk-profile between the groups was the same. Due to the low prevalence of *A. lumbricoides* in both groups, Fisher's exact test [50] was used instead to compare; a p-value of 0.974 allowed us to again assume there was no difference in the risk-profile between the two groups.

## Model fitting

Fig 5 shows the estimates of the regression coefficient for the covariates of M1, M2 and M3 on the odds-ratio scale. We note that the coefficients change depending on the species - providing justification for the interaction terms in the model. For instance, in M1, the effect of age is insignificant for *A. lumbricoides* infections yet has a significant negative effect on the odds for hookworm and *T. trichiura* infections. This heterogeneity of coefficients across species can also be seen in sex in M1, and in all country-level covariates in M2 and M3. The values of the intercept terms and random effect variances are reported in S1 Table.

## Predictive performance

**Overall assessment.** The training dataset comprised 3,818 individuals from 52 countries, while the test set consisted of 52 individuals, each with outcomes relating to three STH species.

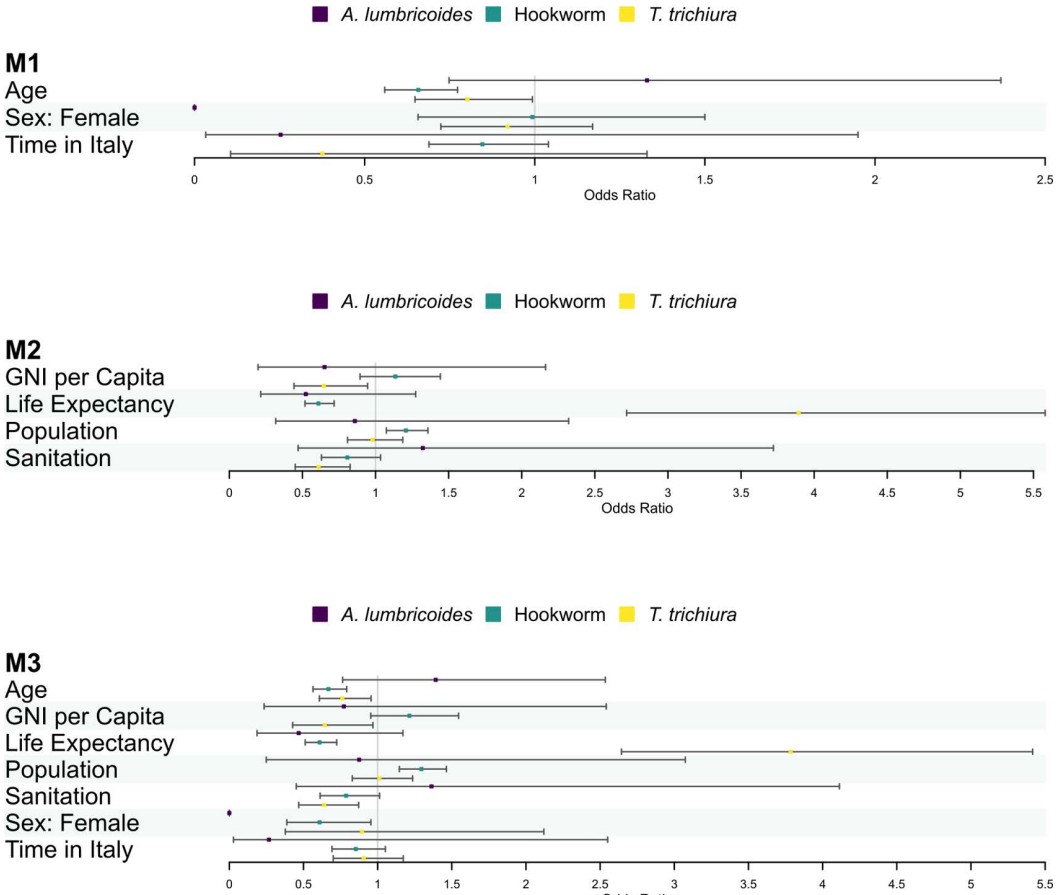

**Fig 5. The estimates and 95% confidence intervals for the fixed effect coefficients in three mixed-effects binomial models.** The three models are: M1 (including only individual-level variables), M2 (including only country-level indicators), and M3 (including both individual-level variables and country-level indicators). Coefficients are on the odds-ratio scale. Each covariate has a coefficient for each species; *A. lumbricoides* in purple, hookworms in green, and *T. trichiura* in yellow.

Fig 6 shows the ROC curves for the prediction of any STH infections under two scenarios, namely for individuals from new countries and for individuals from existing countries. Also shown are the AUC values and 95% DeLong CIs and pairwise DeLong tests for the models' AUCs.

When predicting for individuals from both existing countries and from new countries, M3 performs best, followed by M2 and then M1. Pairwise, the difference in the AUCs is significant for M1, M2 and M3. Both scenarios highlight the importance of considering both country-level indicators and individual-level variables in the prediction of infection.

**Assessment by STH species.** Based on the results shown in Fig 7, we described the results on the predictive performance of the models under the two scenarios considered for each STH species separately.

- *A.* lumbricoides

When predicting for individuals from both existing and new countries, M3 has statistically significantly better AUC than both M1 and M2 but had no significant difference in the AUC of M1 and M2. This shows that utilizing a combination of both country-level indicators and individual-level variables is more effective than a model based on either alone.

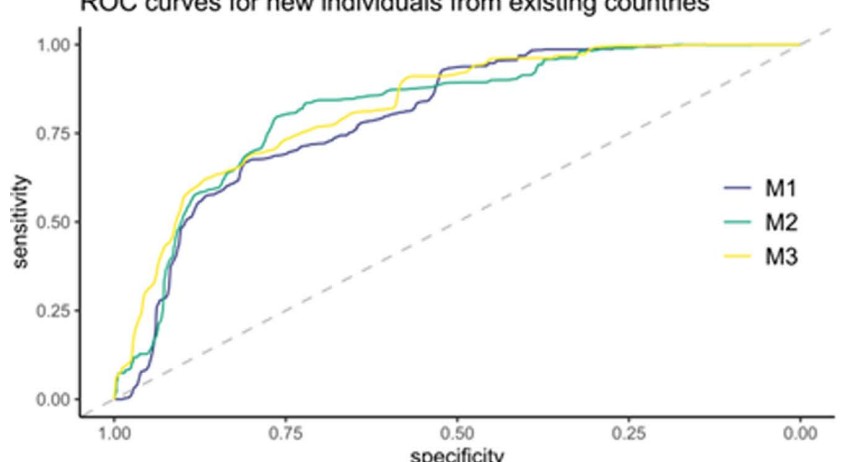

**Fig 6. ROC curves showing the predictive power of three models in two scenarios for predicting STH infection.** The three models are: M1 (utilizing only individual-level variables) shown in purple, M2 (utilizing only country-level indicators) shown in green, and M3 (utilizing both individual-level variables and country-level indicators) shown in yellow. The two scenarios are: Predicting for an individual from one of the 52 countries in the existing dataset (top) and predicting for individuals from new countries not in the original dataset (bottom). The grey lines show the random classifier. To the right of each plot is a table showing the value of the AUC and the DeLong 95% CI for each model, below that is a table showing the p-value for the DeLong test of difference between each pair of models' AUC.

- Hookworms

When predicting for an individual from both existing countries and from new countries, M3 performs the best, followed by M2 and then M1 with each pairwise comparison of the AUCs being statistically significant. This shows that the country-level indicators are more valuable than the individual-level variables when considered alone but that the combination of the two produces the best predictive model in this scenario.

- T. trichiura

When predicting for existing countries, the AUC value for M2 is not significantly better than a random classifier. The AUC of M1 is significantly better than that of M2 and M3, and the AUC of M3 is significantly better than M2. This allows us to

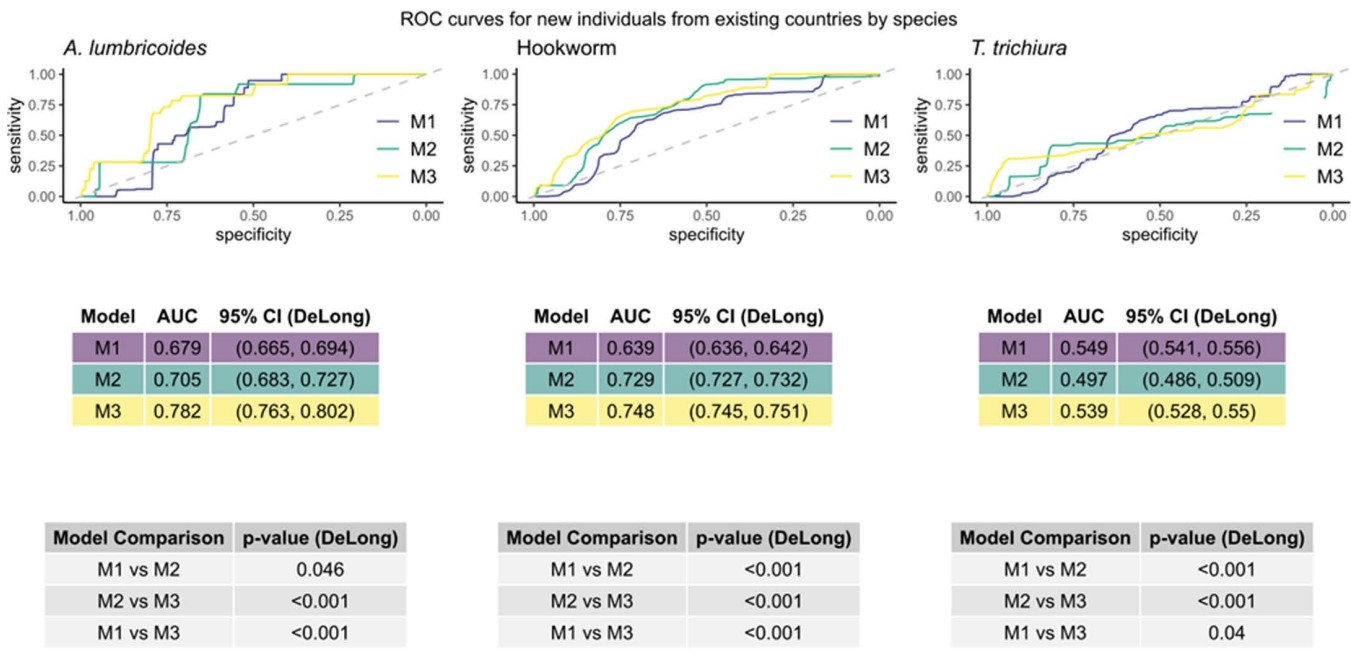

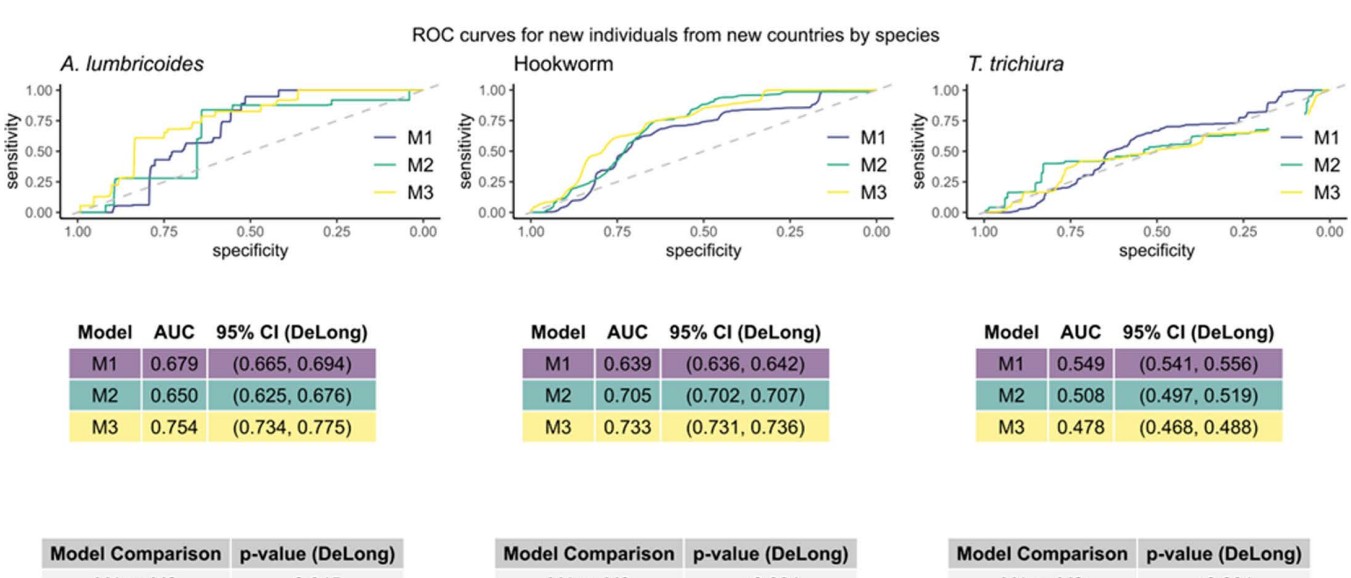

**Fig 7. ROC curves showing the predictive power of three models in two scenarios for predicting STH infection split by each species.** The three species are *A. lumbricoides*, hookworms, and *T. trichiura* (left-to-right). The three models are: M1 (utilizing only individual-level variables) shown in purple, M2 (utilizing only country-level indicators) shown in green, and M3 (utilizing both individual-level variables and country-level indicators) shown in yellow. The two scenarios are: Predicting for an individual from one of the 52 countries in the existing dataset (top) and predicting for individuals from new countries not in the original dataset (bottom). The gray lines show the random classifier. The value of the AUC for each model and the DeLong 95% CI is shown in a table below each ROC plot, beneath that is a table showing the p-value for the DeLong test of difference between each pair of models' AUC.

conclude that in this scenario there is no benefit to including the country-level indicators, and the best model is the one which accounts only for individual-level variables. When predicting for new countries, only M1 performs significantly better than a random classifier, and we draw the same conclusions as for the existing countries when we compare the AUC values pairwise.

### Extended models based on worm indices

Table 1 shows the values of the variance for the random effects in M3, when fitted to the original dataset, and when fitted to a subset of the data for which the worm indices are available, $M4_{WHO}$, $M4_{GBD}$, $M5_{WHO}$, and $M5_{GBD}$.

The 7 models are: M3 fitted to the original dataset, $M4_{WHO}$ and $M4_{GBD}$ which are extensions to M3 considering if a country is in the top 100 countries ranked by the WHO Worm Index and GBD Worm Index respectively, M3 fitted to the subset of data for which the WHO Worm Index is available, $M5_{WHO}$ which is an extension to M3 which considers the rank of the country based on the WHO Worm Index, M3 fitted to the subset of data for which the GBD Worm Index is available, and $M5_{GBD}$ which is an extension to M3 which considers the rank of the country based on the GBD Worm Index.

From this table, we can see that the model which includes whether the individual's country of origin is within the top 100 countries ordered by the WHO Worm Index ($M4_{WHO}$) has a slightly smaller country-level random effect variance than the model which doesn't include this information (M3 fitted to original data). This suggests that the inclusion of this information is accounting for some of the between country-variance which could not be accounted for in M3. The same cannot be said for the model which includes information based on the GBD Worm Index ($M4_{GBD}$), for which the country-level variance is larger than for M3. In both cases, the individual-level random effect variance is larger in the updated model than in M3.

When considering the subset of data for which the WHO Worm Index rank is available, the model which accounts for the rank ($M5_{WHO}$) has a smaller country-level and individual-level random effect than the model which doesn't (M3 fitted to WHO subset). This suggests that the inclusion of the WHO Worm Index rank is improving the model and accounting for more variance than the model not utilizing this information.

Comparing the models fitted to the subset of data for which the GBD Worm Index rank is available, we observe that the country-level random effect variance is smaller in the updated model ($M5_{GBD}$), yet the individual-level random effect is larger. Again, this suggests that some additional country-level variance is being accounted for but at a cost of a larger individual-level random effect variance.

Fig 8 helps us to assess whether the inclusion of this additional information regarding worm indices, and the change in the random-effect variance, influences the predictive performance of the models.

When making predictions about new individuals from existing countries, we observe that M3 has a significantly larger AUC than either $M4_{WHO}$ or $M4_{GBD}$ allowing us to conclude that in this scenario the inclusion of the information regarding whether a country is within the top 100 countries ranked by either Worm Index does not improve the model in terms of prediction.

**Table 1. The value of the country-level and individual-level random effect variances for 7 models.**

| Model | Country-level random effect variance | Individual-level random effect variance |
|---|---|---|
| M3 fitted to original data | 0.227 | 0.0188 |
| $M4_{WHO}$ | 0.214 | 0.117 |
| $M4_{GBD}$ | 0.248 | 0.123 |
| M3 fitted to WHO subset | 0.228 | 0.144 |
| $M5_{WHO}$ | 0.215 | 0.121 |
| M3 fitted to GBD subset | 0.259 | 0.147 |
| $M5_{GBD}$ | 0.238 | 0.199 |

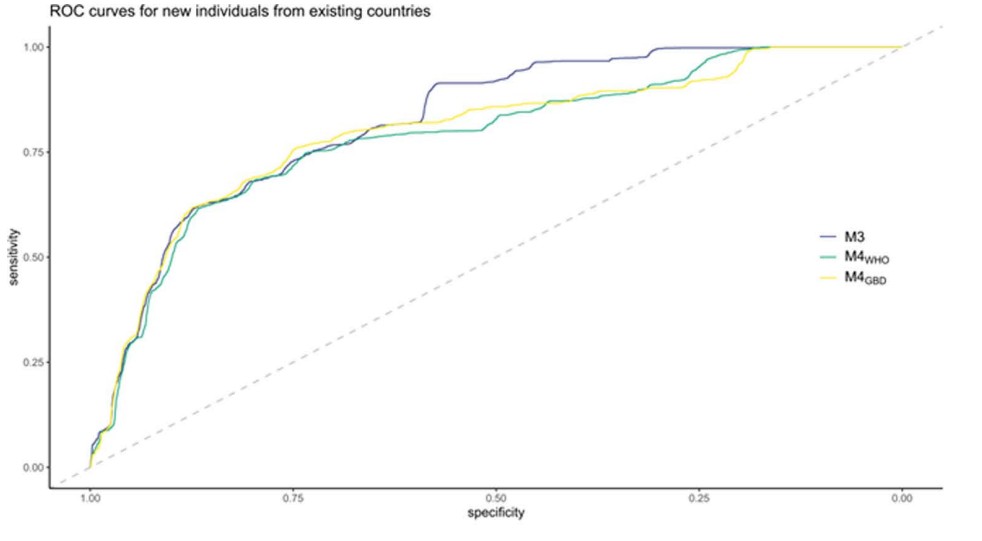

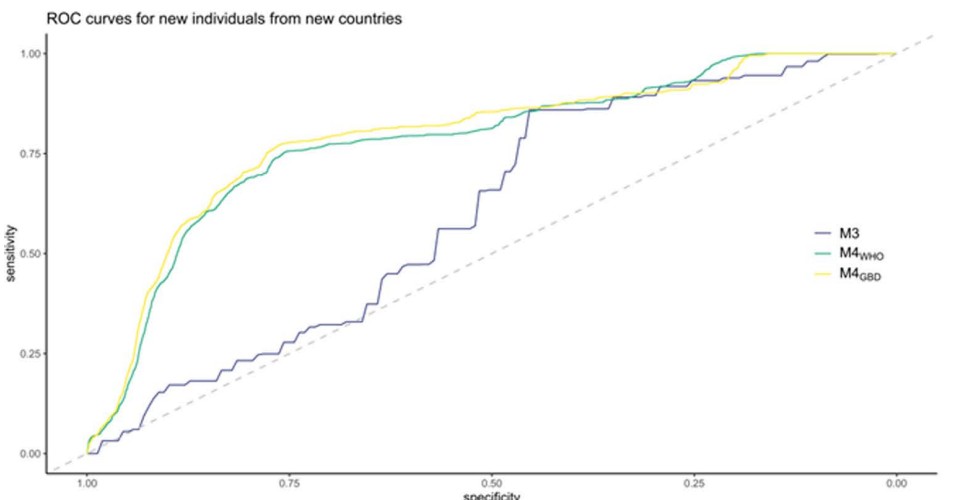

**Fig 8. ROC curves showing the predictive power of three models in two scenarios for predicting STH infections.** The three models are: M3 (utilizing both individual-level variables and country-level indicators) shown in purple, M4$_{WHO}$ and M4$_{GBD}$ shown in green and yellow, respectively. M4$_{WHO}$ and M4$_{GBD}$ are extensions to M3 considering if a country is in the top 100 countries ranked by WHO Worm Index and GBD Worm Index respectively. The two scenarios are: Predicting for an individual from one of the 52 countries in the existing dataset (top) and predicting for individuals from new countries not in the original dataset (bottom). The grey lines show the random classifier. To the right of each plot is a table showing the value of the AUC and the DeLong 95% CI for each model, below that is a table showing the p-value for the DeLong test of difference between each pair of models' AUC.

Assessing the performance of the same models under the scenario where we predict for individuals from new countries, suggests that the inclusion of the additional information is associated with an improvement in model prediction. We conclude this since model M4$_{WHO}$ and M4$_{GBD}$ have significantly larger AUCs than M3.

We also wanted to assess the difference in performance of M3 fitted to WHO subset and M5$_{WHO}$, and between M3 fitted to GBD subset and M5$_{GBD}$ but due to the reduction in the amount of data in these subsets, the models failed to fit the training data. The training sets contained data from only 38 and 36 individuals for the WHO and GBD subsets respectively.

## Discussion

In this study, which focused on the development of predictive models for the likelihood of infection with STHs among a subgroup of the migrant population in southern Italy, we showed how combining individual-level variables and country-level indicators enhances the predictive power of the models. The findings emphasised how the use of a model-based approach could play a key role in driving targeted diagnostic screening on migrants serve as a framework to better inform decisions regarding the treatment of migrants with respect to STH infections.

The STH infections have recently been recognised by the WHO as a major global health problem and prioritized in the WHO's Roadmap for 2030 to eliminate or control worldwide [3]. Given this, further effort is required to reduce the STH prevalence and morbidity in affected populations, considering also refugee and migrant populations to the at-risk groups [51]. Thus, it is of paramount importance to activate screening programs in these populations to guide policies for preventive programs. In this context, the WHO CC ITA-116 (https://ccita116.parassitologia.unina.it/) carries out regular parasitological surveillance and health care for migrants in southern Italy [30] beyond supporting endemic countries in monitoring preventive chemotherapy programmes [52]. By presenting the case study on migrants in southern Italy, we investigated the power of the models in predicting overall STH infections (*A. lumbricoides*, hookworms, and *T. trichiura*) in two main scenarios: for individuals from existing and from new countries. We conclude from our study that in all prediction scenarios, except for predicting *T. trichiura* infections, the best model includes both individual-level variables and country-level indicators, and that the country-level indicators are a stronger predictor than the individual-level for both *A. lumbricoides* and overall STH infections.

We used the AUC and DeLong test to assess the models' performance. It is important to highlight that to use the models on the field, a threshold for the probability of STH infection needs to be identified. Threshold selection poses an additional challenge which must be considered carefully by policy makers and epidemiologists to find the best balance between false-negatives and false-positives. There are many mathematical approaches for finding an optimal threshold [53–57]. The thresholds and corresponding sensitivities and specificities for one of these approaches; the smallest Euclidean distance between the point [0,1] and the ROC curve [53], can be found in S3 Table.

A previous study [58] concluded no evidence of familial transmission within the host country and noted a decline in parasites found in stool samples over time, except for *A. lumbricoides*. Based on this information, we would have expected that the variable indicating the time spent in Italy to be a stronger predictor of infection than we found [59]. Other studies, however, have found that certain helminth species can persist in both the human body [14,60] and the environment [60] for prolonged periods, thus making the length of time since migration a less strong predictor of infection status.

The random effects in the model show that there exists both individual- and country-level variance in the models which cannot be accounted for by the covariates. When making predictions for existing countries, the country-level random effect can be used in place of this unaccounted variance to inform the prediction but the same cannot be said for the individual-level random effects. The size of the variance of the individual-level random effects in M1, the model containing only individual-level variables, is larger than in M3, the model containing both individual-level variables and country-level indicators. This suggests that some of the variance across individuals has been accounted for by the country-level covariates. However, when we assess the fit of the models to the original dataset (see S3 Fig), M1 is the best fitting. This suggests that to improve the model, it would be advisable to incorporate more individual-level variables but in lieu of that information, utilizing country-level indicators is a suitable alternative.

Further evidence that the individual-level variables may be more informative than the country-level indicators comes from the realization that for some groups of migrants, due to the selection pressures of migration, tend to be different from those in their country of origin who did not migrate. Domnich et al. suggest that those migrants may be more educated, and less-risk exposed [61]. This theory follows on from the 'healthy migrant' phenomenon; a phenomenon in which migrants have better health than native-born populations upon migration [61]. This suggests that individuals may not be very well characterized by their country of origin's summary statistics and that if more individual-level variables were

collected it may be more informative than the country-level indicators used in this modelling. Among refugees, a subset of migrants, this 'healthy migrant' phenomenon is not seen, and in fact, they present health deficits due to living conditions in refugee camps [61]. It is documented that in some refugee camps, the sanitation is poor and there exists a lack of access to clean water [62] Based upon our knowledge of STHs (3), it is likely these conditions could make STH infections more prevalent among refugees. A similar study also concluded that the prevalence of NTDs among migrants is likely due to the presence of disease in their country of origin and due to further exposure during migration [12]. It is likely that this model could be improved by considering the reason for migration, and details of an individual's migration journey. In lieu of this information, our model has shown that country-level indicators can act as a suitable predictor.

In addressing the issue of how well we can accurately characterize migrants based on summary statistics from their countries of origin, we should also consider whether our results can be generalized to the broader migrant population. A similar study [12] suggested that those who access health care are more likely to have been in the country for a longer amount of time and have different socio-demographic characteristics, and hence different risk profiles for infection, from the wider migrant population. Italian law states that undocumented migrants have the right to healthcare without being reported to immigration authorities, however in other countries civil servants (including healthcare workers) are legally obliged to report undocumented migrants [63]. This must be considered if this framework were to be applied elsewhere since it might lead those most at risk to be excluded from the screening. Despite the law in Italy, migrants can struggle to access healthcare due to linguistic, cultural and administrative problems [64]. Selection bias cannot be ruled out in this study since enrolment was performed by the healthcare staff, and migrants who did not give their consent to be in the study was not recorded. Due to this bias, we cannot be sure how well the population in our sample represents the overall migrant population in southern Italy and therefore how generalisable the results are.

A previous study conducted by Martelli et al. [12] studied the seroprevalence of five NTDs (Chagas disease, filariasis, schistosomiasis, strongyloidiasis and toxocariasis) in migrant populations in 5 Italian cities (Bologna, Brescia, Florence, Rome, Verona). The inclusion criteria were similar to those of our study, however, Martelli [12] reported a 1:1 male-female ratio, a median age of 37.8 and that almost half of the sample had been living in Italy for more than 10 years. The same summaries for our population sample are substantially different: the male to female ratio is 7:1, the median age is 27 and only 0.05% of people had been in Italy longer than 10 years. Although the etiology of NTDs investigated in Martelli et al. [12] was different from the STHs, some of the results reported in that study show similarities with those reported here. The authors found that age and time since arrival in Italy were not associated with the presence of infections. The present study also found this to be the case true except for hookworms and *T. trichiura* with age. Martelli et al. [12] also found that males were more likely to be infected than females, this is mirrored in our results, with the odds-ratios associated with being female having a value less than 1 for *A. lumbricoides* infections in M1 and both *A. lumbridoides* and hookworm infections in M3. Martelli [12] suggested the reason for this could be due to a higher environmental and working exposure risk in males. Given that Martelli et al.'s study [12] was also conducted in Italy and has such significant differences in sample populations, we can reasonably argue that migrant populations in cities in different countries are likely to be drastically different from the population in this study and therefore the model would require recalibration for other populations.

Whilst the country-level indicators can act as a good predictor, it is known that these country-wide summaries do not account for the so-called "blue marble health", a term used to denote high rates of poverty-related diseases among the poor living in wealthy countries [65]. It has been found that more than half of the world's NTDs and other poverty-related diseases occur in high- and middle-income countries [65] suggesting that country-level covariates may not necessarily be reliable proxies for the likelihood for STH infections in some countries. In investigating this further, models considering information about worm indices had smaller country-level random effect variances, suggesting some of the country-level variation had been accounted for by this addition. However, due to the limited availability of this variable for other countries, we were unable to reliably assess the predictive performance of these models. However, our results suggest that using worm indices may improve the models and aid in better identifying the STH infection status among migrants. Using

the actual value of a country's Worm Index may also prove more informative than the country's ranked position with respect to Worm Index, but this data was not easily available.

We must consider the social implications of this research. Upon investigating the role of international migration on the epidemiology of Chagas disease, Schmunis [66] highlights that because of the findings, it may be necessary to change legislation so that migrants are protected from discrimination due to their infection or potential infection. This seems particularly relevant to this study since countries of origin are being used as predictors. "Community awareness strategies aimed at promoting understanding of the health concerns of refugee populations and countering negative perceptions" are part of the UN Refugee Agency's guidance for resettlement programmes [67]. If the results of this study were used to inform presumptive anthelmintic therapy for those predicted to be at high risk, the potential negatives of this must be considered. These include under-treatment of other NTDs, and the risk of focusing on a single medical intervention whilst neglecting a more comprehensive approach to migrants' health [12,27]. Presumptive anthelmintic therapy should also be accompanied by education measures since non-compliance can be an issue especially among adults who do not believe they are infected unless they can see worms in their stools [60]. It has also been documented that some countries operate pre-departure presumptive treatment for certain groups of migrants [60], which should also be considered in combination with the likelihoods gleaned from the models.

The presence of STH in non-endemic countries is a part of global health and to meet the WHO 2030 goals [68], infections among migrants must also be considered. The methodology presented in this paper can help draw attention especially to at-risk groups such as refugees who otherwise may be excluded from treatment and services due to their mobility and marginalization [69]. However, care must be taken to contextualize this research so as not to divert focus away from the world's poorest populations most burdened by NTDs [70] and to minimize xenophobia and discrimination. On a further note, about the contextualisation of this paper, due to the methodology used in fitting the models to combat computational limitations, the results must only be used for prediction not inference.

## Conclusion

Here we presented a replicable framework to create predictive models for the likelihood of STH infections among migrant populations which can help to make statistically informed decisions. We have shown that the inclusion of both individual-level variables and country-level indicators in the models is important in maximizing the predictive power but also that the most suitable model depends on the modelling scenario and aims of the prediction. The results of such models should be used with caution, acknowledging that the generalizability may be low and the models may require recalibration as spatial-temporal patterns in migration change.

## Supporting information

**S1 File. Further information on the technical details of the binomial mixed-effects models.**
(PDF)

**S2 File. Details of the Delong tests used for model comparison.**
(PDF)

**S1 Fig. The components of the Human Development Index (HDI).** Source [19].
(TIFF)

**S2 Fig. Maps to show the prevalence (%) of STH infection, by species, of migrants by their country of origin split by WHO region. Made using Natural Earth** a) *A. lumbricoides,* b) hookworm and *c) T. trichiura.*
(TIFF)

**S3 Fig. ROC curves for the fitted models for overall infection likelihood and for each species separately.** The yellow lines show the ROC for the full model, the green shows the country-only model, and the purple shows the individual-only model. AUC values and 95% CI are provided for each model in a table to the right of the plot alongside a table for each pairwise comparison of the models.
(TIFF)

**S1 Table. The variance for the random effects terms in models M1, M2, and M3.**
(PDF)

**S2 Table. The values and 95% CIs of the intercept terms in models M1, M2, and M3.**
(PDF)

**S3 Table. The optimal thresholds and corresponding sensitivity (sens) and specificity (spec) for each model in each scenario.** The thresholds are based on the Euclidean distance from (0,1).
(PDF)

## Author contributions

**Conceptualization:** Jana Purkiss, Luciano Gualdieri, Laura Rinaldi, Emanuele Giorgi.

**Data curation:** Naím Alex Karol Poplawski, Maria Paola Maurelli.

**Formal analysis:** Jana Purkiss.

**Investigation:** Paola Pepe, Maria Paola Maurelli.

**Methodology:** Jana Purkiss, Emanuele Giorgi.

**Resources:** Luciano Gualdieri.

**Supervision:** Laura Rinaldi, Emanuele Giorgi.

**Visualization:** Jana Purkiss, Naím Alex Karol Poplawski.

**Writing – original draft:** Jana Purkiss, Paola Pepe, Emanuele Giorgi.

**Writing – review & editing:** Laura Rinaldi.

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
