## [Decision Letter · Decision Letter 0]

29 Jan 2025

Dear Dr. Purkiss,

Response to Reviewers
Revised Manuscript with Track Changes
Manuscript

Shaden Kamhawi

co-Editor-in-Chief

Paul Brindley

co-Editor-in-Chief

**Additional Editor Comments:**

Please carefully address all the reviewers' comments herein below.

**Journal Requirements:**

1) Please provide an Author Summary. This should appear in your manuscript between the Abstract (if applicable) and the Introduction, and should be 150-200 words long. The aim should be to make your findings accessible to a wide audience that includes both scientists and non-scientists. Sample summaries can be found on our website under Submission Guidelines:

2) Some material included in your submission may be copyrighted. According to PLOSu2019s copyright policy, authors who use figures or other material (e.g., graphics, clipart, maps) from another author or copyright holder must demonstrate or obtain permission to publish this material under the Creative Commons Attribution 4.0 International (CC BY 4.0) License used by PLOS journals. Please closely review the details of PLOSu2019s copyright requirements here: PLOS Licenses and Copyright. If you need to request permissions from a copyright holder, you may use PLOS's Copyright Content Permission form.

Potential Copyright Issues:

- Figures 4 and S2; Please provide a direct link to the base layer of the map (i.e., the country or region border shape) and ensure this is also included in the figure legend; and provide a link to the terms of use / license information for the base layer image or shapefile. We cannot publish proprietary or copyrighted maps (e.g. Google Maps, Mapquest) and the terms of use for your map base layer must be compatible with our CC BY 4.0 license.

3) Please amend your detailed Financial Disclosure statement. This is published with the article. It must therefore be completed in full sentences and contain the exact wording you wish to be published.

**Reviewers' comments:**

**Key Review Criteria Required for Acceptance?**

**Methods**

-Are the objectives of the study clearly articulated with a clear testable hypothesis stated?

-Is the study design appropriate to address the stated objectives?

-Is the population clearly described and appropriate for the hypothesis being tested?

-Is the sample size sufficient to ensure adequate power to address the hypothesis being tested?

-Were correct statistical analysis used to support conclusions?

-Are there concerns about ethical or regulatory requirements being met?

Reviewer #1: All of the above points have been correctly considered by the authors.

Reviewer #2: Yes

**Results**

-Does the analysis presented match the analysis plan?

-Are the results clearly and completely presented?

-Are the figures (Tables, Images) of sufficient quality for clarity?

Reviewer #1: Results, figures and tables have been clearly presented.

Reviewer #2: Yes. However results could be better laid out to emphasise the predictive value of the models.

**Conclusions**

-Are the conclusions supported by the data presented?

-Are the limitations of analysis clearly described?

-Do the authors discuss how these data can be helpful to advance our understanding of the topic under study?

-Is public health relevance addressed?

Reviewer #1: The conclusions have been properly discussed and the the subject studied is of public health concern.

Reviewer #2: Yes, as authors mentioned special attention and caution should be taken with these type of studies giving the proper contextualisation. Authors stated this is important to minimise xenophobia and discrimination, I should suggest to use a different term such as to avoid.

**Editorial and Data Presentation Modifications?**

Reviewer #1: The manuscript is well-organized and well-written, and the issue is very topical and of great interest. I personally suggest just few minor corrections.

In the Materials and Methods, at line 101 “patients were not taking drugs or other substances”, specify how long ago and what “substances” you are referring to; at line 111, indicate that one stool sample for patient was analyzed.

In the results section, I suggest to briefly report some more data on the prevalence of STHs in migrants coming from countries of Africa and South-East Asia indicated in yellow in Fig. 4. Please report the results of copro-parasitological examinations by using the two flotation solutions.

Line 541, change “were” into “was”.

Line 545, change “the number of migrants who did not give their consent to be in the study was not recorded.” into “the migrants who did not give their consent to be in the study were not recorded.”

Lines 555-556, change “Although the species of NTDs investigated in Martelli et al. were different from…..” into “Although the etiology of NTDs investigated in Martelli et al. was different from…….”.

Reviewer #2: Line 38 - 39. is there any evidence of emergence of NTD in developed countries?

Line 54. please ad a reference.

Line 60. please ad a reference.

Lines 89- 90. the objective is not very clear please modify also using third person

line 92. not clear either or both

Line 107. delete since they arrive in Italy

Line 141. yet that

Line 145 , before respectively.

Line 172. when available?

Line 174. making

Line 203. delete "by the" (before mean)

Line 306. three

Line 348 - 352. this paragraph seems repeated from the methods.

Line 373 we described

Line 378 but had

Line 467. The first paragraph of the discussion should be the main finding related to the main objective.

Line 467. . instead of ;

Line 487 field,

line 494 [53]

Line 551. were

Line 556. in that study

Line 558. the present study

Line 562. suggested

Line 574 - 580 this seems repeated from methods and results. delete of modify

Line 585. using or utilising

Line 589. delete another NTD

Line 592. it is not clear (for example because of this study given??) please modify

Line 615. here we presented

**Summary and General Comments**

Reviewer #1: (No Response)

Reviewer #2: The manuscript PNTD-D-24-01402 Combining country indicators and individual variables to predict soil-transmitted helminth infections among migrant populations: a case study from southern Italy, is overall well written and uses many statistical and predictive model approaches to assess STH in migrant populations. in this case using southern Italy as a case study. As authors mentioned, the narrative should be towards the predictive value of such models and the contextualisation of each particular case. This is pivotal to avoid more discrimination and xenophobia, given also the present polarised political climate. With that said authors highlighted this possibility accurately and the paper does reflect the opportunity of using this models to predict STH, rather than inferring.

PLOS authors have the option to publish the peer review history of their article (what does this mean? ). If published, this will include your full peer review and any attached files.

**Do you want your identity to be public for this peer review?** For information about this choice, including consent withdrawal, please see our Privacy Policy .

Reviewer #1: No

Reviewer #2: No

**Figure resubmission:****Reproducibility:** To enhance the reproducibility of your results, we recommend that authors of applicable studies deposit laboratory protocols in protocols.io, where a protocol can be assigned its own identifier (DOI) such that it can be cited independently in the future. Additionally, PLOS ONE offers an option to publish peer-reviewed clinical study protocols. Read more information on sharing protocols at https://plos.org/protocols?utm_medium=editorial-email&utm_source=authorletters&utm_campaign=protocols

---

## [Editor Report · Decision Letter 1]

13 May 2025

Dear Purkiss,

We are pleased to inform you that your manuscript 'Combining country indicators and individual variables to predict soil-transmitted helminth infections among migrant populations: a case study from southern Italy.' has been provisionally accepted for publication in PLOS Neglected Tropical Diseases.

Best regards,

Domenico Otranto

Academic Editor

Justin Remais

Section Editor

Shaden Kamhawi

co-Editor-in-Chief

Paul Brindley

co-Editor-in-Chief

The Authors have correctly addressed the comments and criticisms raised by the reviewers.

---

## [Editor Report · Acceptance letter]

Dear Purkiss,

We are delighted to inform you that your manuscript, "Combining country indicators and individual variables to predict soil-transmitted helminth infections among migrant populations: a case study from southern Italy.," has been formally accepted for publication in PLOS Neglected Tropical Diseases.

Best regards,

Shaden Kamhawi

co-Editor-in-Chief

Paul Brindley

co-Editor-in-Chief
